# Evaluating the effectiveness of preservice midwifery curricula in Ethiopia: A comparison of neonatal resuscitation and infection prevention practice of midwifery graduates trained in competency-based versus conventional curricula

Awoke Giletew Wondie[1,2]*, Matthias Siebeck[1,3], Tegbar Yigzaw Sendekie[4], Martin Fischer[3], Markus Berndt[3]

1 CIH LMU, Center for International Health, LMU University Hospital, LMU Munich, Munich, Germany, 2 Department of Reproductive and Child Health, College of Health Sciences, Debre Tabor University, Debre Tabor, Ethiopia, 3 Institute of Medical Education, LMU University Hospital, LMU Munich, Munich, Germany, 4 Jhpiego (an affiliate of John Hopkins University), Addis Abeba, Ethiopia

* awoke.wondie@lrz.uni-muenchen.de, awokegiletew@yahoo.com

## Abstract

### Background

Infection control and neonatal resuscitation are essential midwifery practices that can reduce maternal and neonatal mortality. However, in Ethiopia, theory-heavy midwifery education leads to limited clinical competence. To address this, Debre Tabor University implemented a competency-based curriculum in 2013. This study examines whether competency-based midwifery education produces graduates with significantly better performance in neonatal resuscitation and infection prevention compared to conventional education, thereby framing a testable argument about the curriculum's effectiveness.

### Methods

A comparative cross-sectional study assessed the infection prevention and neonatal resuscitation performance of 68 BSc midwifery graduates (32 competency-based vs. 36 conventional) from third-generation Ethiopian universities. Performance was measured using a validated observation tool in clinical settings for infection prevention and simulations for neonatal resuscitation. Mean percentage scores were compared using *t*-tests, with effect size illustrated via Gardner–Altman plots.

### Results

Overall, midwives demonstrated 63.6% of essential neonatal resuscitation tasks, with competency-based curriculum graduates (CBCGs) having higher performance

**Data availability statement:** All relevant data are within the manuscript and its Supporting information files.

**Funding:** This study was supported by Jhpiego's Health Workforce Improvement Project in the form of a grant awarded to TYS and by Jhpiego's Health Workforce Improvement Project in the form of a salary for TYS. The specific roles of this author are articulated in the Author Contributions section. The funders had no role in study design, data collection and analysis, decision to publish, or preparation of the manuscript.

**Competing interests:** A. G. W. and T. Y. S. were involved in the design and implementation of the competency-based curriculum at Debre Tabor University. To minimize bias, these authors did not participate in the data collection, observation, or scoring processes. Data collection was conducted by trained independent midwife observers who were blinded to the graduates' curriculum background. Author T. Y. S. is employed by Jhpiego's Health Workforce Improvement Project. This employment provided salary support but did not influence the study design, data collection and analysis, decision to publish, or preparation of the manuscript. There are no patents, products in development, marketed products, or consultancies related to this research. This does not alter our adherence to PLOS ONE policies on sharing data and materials.

than conventional curriculum graduates (CCGs) (71.6% vs. 56.5%; $t$ (66.0) = 3.82, $p < .001$; difference = 15.1%), particularly in airway suctioning and chest rise assessment. For infection prevention, midwives performed 71.7% of the required tasks, with CBCGs again scoring higher (76.9% vs. 67.0%; $t$ (55.4) = 2.79, $p < .01$; difference = 9.9%). Key differences were observed in hand hygiene, the use of personal protective equipment, and apron decontamination. Despite these improvements, persistent deficiencies remained in both groups, particularly in checking breathing/pulse during neonatal resuscitation and in disinfecting aprons during infection prevention practices.

## Conclusions

CBCGs demonstrated better performance in neonatal resuscitation and infection prevention compared to those from the conventional program, suggesting clinical relevance. However, performance gaps in both groups underscore the need for enhanced simulation training, ongoing skill reinforcement, and curriculum refinement.

## 1. Introduction

Infection prevention and neonatal resuscitation are critical components of midwifery care, essential for reducing maternal and neonatal morbidity and mortality in low- and middle-income countries (LMICs) [1]. Midwives play a central role in preventing infections through early detection of maternal infections and by using sterile techniques during labor and delivery. They are also the primary responders when newborns fail to breathe at birth, managing cases with basic interventions such as warming, airway clearance, and stimulation, as well as advanced procedures like bag-and-mask ventilation and chest compressions when necessary. These two competencies are closely linked, as effective infection control can reduce complications, such as sepsis, that may require resuscitation. Together, they contribute to improved maternal and neonatal outcomes and support progress toward Sustainable Development Goal 3 of the United Nations [2,3]. Strengthening midwifery education in these areas is particularly important in resource-limited settings, where skilled care has a direct impact, and curriculum design in preservice education (PSE) plays a pivotal role in developing these life-saving competencies [4].

In Ethiopia, traditional theory-heavy midwifery education has resulted in graduates with limited clinical competence, particularly in essential life-saving procedures. To address these gaps, Debre Tabor University (DTU) implemented a competency-based integrated curriculum in 2013, explicitly designed to translate theoretical knowledge into practical skills through problem-based learning, early clinical exposure, simulation-based training, and community-based education. This curriculum represents a targeted response to the deficiencies of conventional programs, directly linking educational reform to improved clinical performance [5].

While evidence highlights the benefits of active, community-oriented education in shaping empathy, diagnostic abilities, and communication skills, few studies have

examined how curriculum design in PSE influences intermediate health outcomes [5,6]. This study investigates whether such curriculum reform enhances the provision of neonatal resuscitation and infection prevention services by comparing graduates from competency-based and conventional programs. By focusing on these lifesaving procedures as an inter-mediate outcome, the study addresses a critical evidence gap linking PSE to clinical practice quality and aims to inform global efforts to improve maternal and neonatal care.

This study tests the hypothesis that graduates of competency-based midwifery curricula demonstrate superior perfor-mance in neonatal resuscitation and infection prevention compared to those trained under conventional curricula, thereby framing a clear, debatable argument about the curriculum's effectiveness in preparing clinically competent midwives.

## 2. Methods

### 2.1. Study design

A comparative cross-sectional study assessed whether a competency-based PSE curriculum led to improved clinical per-formance of midwifery graduates.

**Setting.** This study was conducted in Ethiopia, where midwifery education has traditionally relied on a theory-heavy, conventional curriculum with limited practical training. In 2013, DTU introduced a competency-based curriculum, guided by the SPICES model and tailored to Ethiopia's educational and healthcare needs [5,6]. While early evaluations suggested positive academic outcomes [5], recently, the curriculum has been adopted nationally without a rigorous assessment of its impact on health outcomes. To address this evidence gap, the study compared the performance of DTU graduates with peers from third-generation public universities with similar contexts. Data were collected from August 26 to October 12, 2024.

### 2.2. Population

The study targeted recently deployed Bachelor of Science (BSc) midwifery graduates from third-generation Ethiopian universities. The inclusion criteria were midwifery professionals who passed the national licensure exam, were employed within one year after graduation, had served for not more than one year, and were assigned to childbirth care.

The study focused on eligible midwives, not the women observed. Cases involving other providers or procedures beyond midwives' scope (e.g., cesarean deliveries) were excluded.

### 2.3. Sample size and selection of study participants

The number of participants was calculated based on a power analysis to determine a moderate effect size (d = 0.5) with 80% power, a 95% confidence level and α = .05, ensuring sufficient power to detect a medium-to-large effect size between groups, resulting in 32 participants per group. Of 37 recent DTU graduates, 32 met the inclusion criteria; five were excluded due to unemployment, career change, or working outside the study area. Three of the nine third-generation universities (excluding DTU) were excluded because two had already started implementing similar curricula, and one lacked sufficient data. To account for anticipated non-response, a 10% margin of error was added, resulting in a final sample of 36 randomly selected participants from the comparison group's registrar lists. Participants were employed across all regions of Ethiopia, and the inclusion of both public hospitals and health centers serving urban and rural popula-tions enhances contextual representativeness.

### 2.4. Data collection procedures and instruments

The primary outcomes were the performance of neonatal resuscitation and infection prevention practice at birth (**S1 Data**). Data was collected through **direct structured observation** of labor and deliveries to assess infection prevention practices and evaluate midwives' newborn resuscitation skills in a simulated environment using the NeoNatalie model [7].

Adherence to infection prevention practices was measured using a standardized tool from the Maternal, Newborn, and Child Health Integrated Program Quality of Care Assessment for Labor, Delivery, and Newborn Care [8]. It was assessed in detail, focusing on critical components, including proper hand hygiene before and after procedures, the use of sterile gloves, the appropriate use of personal protective equipment (PPE), instrument decontamination, the safe disposal of sharps and waste, and the proper removal and disinfection of aprons. Neonatal resuscitation performance was evaluated through a validated, competency-based simulation assessment of resuscitation skills [9].

Two trained, independent observers assessed participants' performance during simulated neonatal resuscitation and labor and delivery sessions. Each task was scored as 'performed correctly' or 'not performed' to calculate a percentage performance score. Observers underwent a five-day training program that covered study protocols, detailed task scoring criteria, interrater calibration, and role-play exercises to ensure consistency and reliability. Observers were blinded to participants' curriculum group to minimize bias. Average scores from two independent observers were used for analysis, enhancing the reliability and replicability of the evaluation.

**Data collection field work.** Study procedures (detailed in **S1 Fig**) involved screening eligible midwives and inviting laboring women in established labor to participate. Observations began with initial evaluation and continued through childbirth and up to one hour postpartum. Midwives observed while providing labor and delivery service were asked to perform a simulated newborn resuscitation. Two trained observers independently and unobtrusively documented care, using average scores for analysis. Only cases where the primary provider was the selected midwife were included. Data was collected using the Kobo Collect mobile platform.

## 2.5. Data analysis

Data were analysed using IBM Statistical Package for the Social Sciences (SPSS) Statistics, version 27. Performance was measured as the percentage mean score for each outcome (neonatal resuscitation and infection prevention). Independent t-tests were used to compare graduates from competency-based and conventional curricula, with effect sizes (Cohen's d) visualized by Gardner–Altman plots. The assumptions of normality and homogeneity of variance were verified using the Shapiro–Wilk and Levene's tests, respectively. These outcomes were part of four primary outcomes assessed in a broader study. Multiple comparison corrections were applied using Bonferroni, Holm–Bonferroni, and false discovery rate methods to account for related outcomes. Both results remained statistically significant under all adjustments.

## 2.6. Interrater reliability and internal consistency

Although previously validated in a similar setting, the tool's internal consistency was re-evaluated using data from this study. Interrater reliability of the total performance scores was assessed using a two-way mixed effects intraclass correlation coefficient (ICC) with a consistency definition. The ICCs for average measures of infection prevention and neonatal resuscitation were 0.87 (95% CI: 0.79–0.92, $p < 0.001$) and 0.84 (95% CI: 0.74–0.90, $p < .001$), respectively, suggesting excellent agreement between the two raters. Additionally, Cronbach's alpha for infection prevention and neonatal resuscitation items was 0.66 and 0.64, respectively, indicating acceptable internal consistency for the purpose of this study.

## 2.7. Ethical considerations

The study protocol was approved by the Ethical Review Committee of Debre Tabor University, Ethiopia (Date: 13.07.2023/ No: RP/251/23), and the Ethics Commission of the Medical Faculty of Ludwig-Maximilians-Universität, Munich, Germany (Date: 18.10.2023/ Project Nr.: 23–0757). At each health facility, the supervisor presented an official letter from DTU to the facility director requesting cooperation. For midwives, both ethics committees approved the use of oral informed consent. Although written consent is generally preferred, requiring signatures during clinical shifts was determined to risk disrupting urgent intrapartum care. To minimize coercion and ensure voluntariness, midwives were approached individually, outside the immediate care setting, and informed about the study's purpose, risks, and benefits. They were also provided

the opportunity to decline participation without any involvement from facility administrators. Their oral consent was documented by trained data collectors as approved in the ethical protocol.

For laboring women, the ethics committees granted a waiver of written consent. Many women were in active labor or experiencing discomfort, and requesting written signatures, including check marks or symbolic signatures, was considered to impose an additional burden during a vulnerable moment. Instead, data collectors read the consent information aloud in the woman's preferred language, answered questions, and obtained verbal consent, which was recorded in the data collection system. Women were assured that refusal would not affect the care they received. Across all participants, confidentiality was strictly maintained: no identifying information about midwives, women, or facilities was collected, and all results were reported anonymously. All consent procedures were conducted in accordance with approved protocols, with an emphasis on minimizing burden, respecting autonomy, and safeguarding participants' rights.

All informed consent procedures were reviewed and approved by the respective ethics committees.

## 3. Results

### 3.1. Characteristics of participants and health facilities

**Service providers.** A total of 68 midwifery service providers were observed. Most (97.1%) had 6–12 months of experience, aligning with the time since earning their BSc (**Table 1**).

**Client and facility characteristics.** Sixty-eight laboring women were included, most aged 25–34 years (86.7%) and from urban areas (64.7%). All observations occurred in public facilities, primarily health centers (65.6%), with comparable representation from both groups (**S1 Table**).

**Table 1. Characteristics of midwives observed by type of curriculum in preservice education ($N = 68$).**

| Variables | | CCGs [a] ($n = 36$) | | CBCGs [b] ($n = 32$) | | Total ($N = 68$) | |
|---|---|---|---|---|---|---|---|
| | | N | % | n | % | n | % |
| Age<br>Mean $\pm$ SD $= 24.58 \pm 0.814$ | < 25 | 14 | 38.9 | 14 | 43.7 | 28 | 41.2 |
| | 25–29 | 22 | 61.1 | 18 | 56.3 | 40 | 58.8 |
| Sex | Male | 16 | 44.4 | 21 | 65.6 | 37 | 54.4 |
| | Female | 20 | 55.5 | 11 | 30.5 | 31 | 45.6 |
| Marital status | Married | 9 | 25.0 | 6 | 18.7 | 15 | 22.1 |
| | Not married | 27 | 75.0 | 26 | 81.3 | 53 | 77.9 |
| Experience | 0–5 months | 2 | 5.5 | 0 | 0.0 | 2 | 2.9 |
| | 6–12 months | 34 | 94.2 | 32 | 100 | 66 | 97.1 |
| Workplace by region/City administrations | Addis Ababa | 11 | 30.5 | 12 | 37.5 | 23 | 33.8 |
| | Amhara | 14 | 38.9 | 9 | 28.1 | 23 | 33.8 |
| | Dire Dawa | 4 | 11.2 | 3 | 9.4 | 7 | 10.3 |
| | Somali | 2 | 5.5 | 2 | 6.2 | 4 | 5.9 |
| | Central Ethiopia | 4 | 11.1 | 4 | 12.5 | 8 | 11.7 |
| | Southern Ethiopia | 1 | 2.7 | 2 | 6.2 | 3 | 4.4 |
| Type of facility | Referral hospital | 5 | 13.9 | 4 | 12.5 | 9 | 13.2 |
| | General hospital | 6 | 16.7 | 4 | 12.5 | 10 | 14.7 |
| | Primary hospital | 7 | 19.4 | 5 | 15.6 | 12 | 17.7 |
| | Health center | 18 | 50.0 | 19 | 59.4 | 37 | 54.4 |

*a = Conventional curriculum graduates, b = Competency-based curriculum graduates.*

## 3.2. Neonatal resuscitation performance

Overall, midwives' performance in demonstrating neonatal resuscitation was 63.6% (95% CI: 59.4–67.8). Midwives' performance in initial steps (drying/stimulation/suctioning) was consistently better than tasks required for ventilation. This result was higher among competency-based curriculum graduates (CBCGs) (71.6%) compared to the conventional curriculum graduates (CCGs) (56.5%). The key differences were observed in airway suctioning (87.5% vs 66.7%) and chest rise assessment (56.3% vs. 36.1%), while breathing/pulse was checked by less than 47% of midwives in both groups (Table 2).

## 3.3. Infection prevention practice

Overall, midwives implemented 71.7% (95% CI: 68.1–75.3) of infection prevention tasks. Universal glove use and proper waste disposal were consistently observed, highlighting that while basic infection prevention measures are routinely applied, critical components of sterile technique revealed gaps in key practices, including hand hygiene (48.6%), PPE use (83.8%), and apron decontamination (11.8%) (Table 3).

Comparing group performance, CBCGs scored higher (76.9%) than those trained under the conventional curriculum (67%). The key differences were observed in hand hygiene (62.5% vs 36.1%), PPE use (90.6% vs. 77.8%), and apron decontamination (18.7% vs. 5.6) (Table 3).

## 3.4. Comparing performance

An independent samples *t*-test was conducted to compare the mean percentage score for the performance of neonatal resuscitation and infection prevention practice. Results showed statistically significant differences favoring CBCGs in both

**Table 2. Performance of midwives in performing steps of neonatal resuscitation during simulated resuscitation by type of preservice curriculum in Ethiopia (N = 68).**

| Variables | CCGs [a] (n = 36) | | CBCGs [b] (n = 32) | | Total (n = 68) | |
|---|---|---|---|---|---|---|
| | n | % | n | % | n | % |
| **Initial steps (drying/stimulation/suctioning)** | | | | | | |
| Dries the baby and wraps the baby in a dry, warm cloth. | 28 | 77.8 | 28 | 87.5 | 56 | 82.4 |
| Places the baby on their back on a clean, warm surface | 31 | 86.1 | 30 | 93.8 | 61 | 89.7 |
| Positions the head in a slightly extended position | 21 | 58.3 | 24 | 75.0 | 45 | 66.2 |
| Clears the airway | 24 | 66.7 | 28 | 87.5 | 52 | 76.5 |
| Introduces a catheter into the baby's mouth | 22 | 61.1 | 23 | 71.9 | 45 | 66.2 |
| Introduces the catheter into each nostril for approximately 3 cm and suctions while withdrawing the catheter | 20 | 55.6 | 22 | 68.8 | 42 | 61.8 |
| **Ventilation** | | | | | | |
| Places a mask on the baby's face | 21 | 58.3 | 25 | 78.1 | 46 | 67.6 |
| Squeezes the bag with two fingers | 23 | 63.9 | 25 | 78.1 | 48 | 70.6 |
| Checks the seal and observing the chest's rise. | 13 | 36.1 | 18 | 56.3 | 31 | 45.6 |
| Ventilates at a rate of 40 breaths/minute | 17 | 47.2 | 22 | 68.8 | 39 | 57.4 |
| Assesses for spontaneous breathing | 13 | 36.1 | 15 | 46.9 | 28 | 41.2 |
| Checks pulse/heart rate | 11 | 30.6 | 15 | 46.9 | 26 | 38.2 |
| **Mean percentage score for NR[c] (95%CI)** | **56.5** **(51.9–61.1)** | | **71.6** **(65.2–78.0)** | | **63.6** **(59.4–67.8)** | |

*a = Conventional curriculum graduates, b = Competency-based curriculum graduates, c = Neonatal Resuscitation.*

**Table 3. Performance of midwives in infection prevention practices during labor and delivery by a curriculum in preservice education, Ethiopia (*N* = 68).**

| Actions | CCGs [a] (*n* = 36) | | CBCGs [b] (*n* = 32) | | Total (*n* = 68) | |
|---|---|---|---|---|---|---|
| | n | % | n | % | n | % |
| Hand wash before procedure | 13 | 36.1 | 20 | 62.5 | 33 | 48.6 |
| Wears sterile gloves before any vaginal examination | 36 | 100 | 32 | 100 | 68 | 100 |
| Wears clothing to protect face, hands and body | 28 | 77.8 | 29 | 90.6 | 57` | 83.8 |
| Proper sharp disposal | 30 | 83.3 | 28 | 87.5 | 58 | 85.3 |
| Decontaminates instruments | 28 | 77.8 | 24 | 75.0 | 52 | 76.5 |
| Disposes of all contaminated waste in leakproof containers | 36 | 100 | 32 | 100 | 68 | 100 |
| Removes aprons and wipes with chlorine solution | 2 | 5.6 | 6 | 18.7 | 8 | 11.8 |
| Hand wash after procedure | 20 | 55.6 | 26 | 81.3 | 46 | 67.6 |
| **Mean percentage score for infection prevention (95%CI)** | **67.0 (61.0–73.0)** | | **76.9 (73.4–80.4)** | | **71.7 (68.1–75.3)** | |

*a = Conventional curriculum graduates, b = Competency-based curriculum graduates.*

**Table 4. Comparing neonatal resuscitation and infection prevention practice performance by innovative and conventional curriculum midwife graduates, 2024.**

| Outcome | Means | SD | F | Sig | t | df | Sig(2-tailed) | Mean Difference | Std. Error Difference | 95% CI[c] | |
|---|---|---|---|---|---|---|---|---|---|---|---|
| | | | | | | | | | | Lower | Upper |
| **Infection prevention practice** | | | | | | | | | | | |
| **CBCGs[a]** | 0.769 | 0.1009 | 16.31 | 0.000 | 2.706 | 66 | 0.009 | 0.09939 | 0.0367 | 0.026 | 0.172 |
| **CCGs[b]** | 0.670 | 0.1845 | | | **2.795** | **55.43** | **0.007** | **0.09939** | **0.0355** | **0.028** | **0.170** |
| **Neonatal resuscitation performance** | | | | | | | | | | | |
| **CBCGs[a]** | 0.716 | 0.184 | 3.83 | 0.054 | **3.826** | **66** | **0.000** | **0.15133** | **0.03956** | **0.0723** | **0.2303** |
| **CCGs[b]** | 0.564 | 0.141 | | | 3.766 | 57.78 | 0.000 | 0.15133 | 0.04018 | 0.0709 | 0.2317 |

*a = Competency-based curriculum graduates, b = Conventional curriculum graduates, C = Confidence interval.*

neonatal resuscitation (*M* = 0.716 vs. 0.565; *t* (66) = 3.82, 95% CI [0.23, 0.72], *p* < .001) and infection prevention (*M* = 0.769 vs. 0.670; *t* (55.4) = 2.79, 95% CI [0.03, 0.17], *p* < .01) (**Table 4**).

**Effect size estimation.** The variation in neonatal resuscitation performance across groups was estimated using a Gardner–Altman plot generated via the DABEST package [10]. The analysis for neonatal resuscitation revealed a **mean difference (Δ) of 15%** [95% CI:.07 to.23] on a standard 0–1 scale of quality, preferring graduates of the CBCGs (*M* = 0.71, *SD* = .18, *n* = 32) than the CCGs (*M* = .56, *SD* = .14, *n* = 36) (**Fig 1**).

The analysis for infection prevention revealed a **mean difference (Δ) of 9.9%** [95% CI:.03 to.17] on a standard 0–1 scale of quality, preferring graduates of the CBCGs (*M* = 0.77, *SD* = .10, *n* = 32) than the CCGs (*M* = .67, *SD* = .18, *n* = 36) (**Fig 2**).

## 4. Discussion

This study compared the performance of neonatal resuscitation and infection prevention practice by midwifery graduates from conventional and competency-based curricula. Midwives demonstrated 63.6% proficiency in neonatal resuscitation, with CBCGs having higher performance than CCGs (71.6% vs. 56.5%), a statistically and practically significant difference (15% mean difference), particularly in airway suctioning and chest rise assessment. For infection prevention, midwives performed 71.7% of required tasks, with CBCGs again scoring higher (76.9% vs. 67%), showing

 

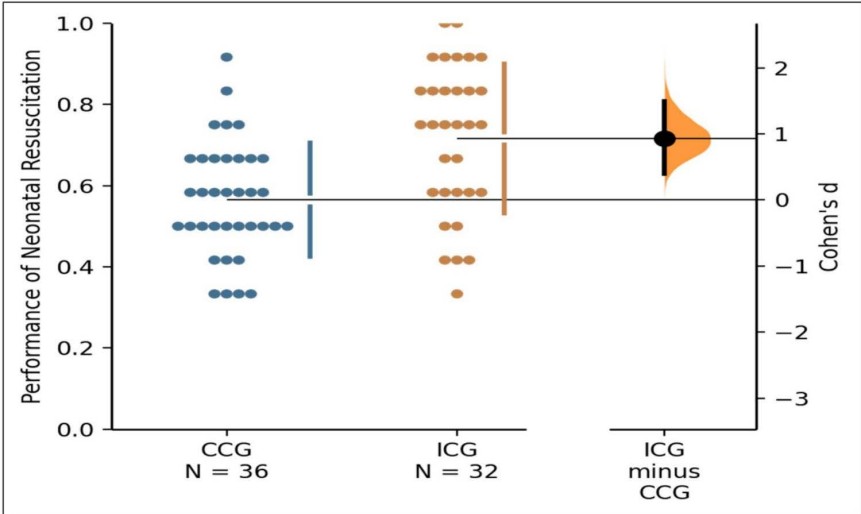

**Fig 1. A Gardner-Altman plot estimating the difference in neonatal resuscitation performance between competency-based and conventional curriculum graduates.**

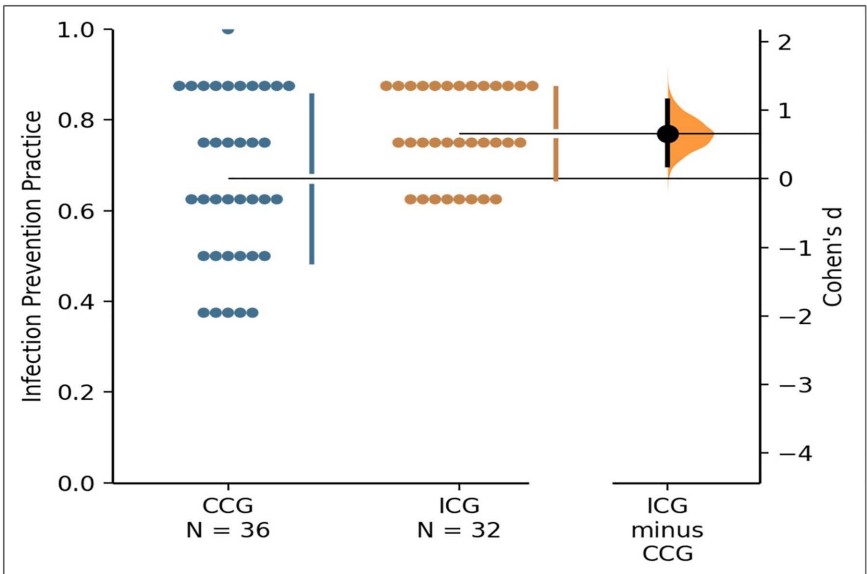

**Fig 2. A Gardner-Altman plot estimating the difference in infection prevention practice between competency-based and conventional curriculum graduates.**

a meaningful 9.9% effect size. Key differences were noted in hand hygiene, PPE use, and apron decontamination. Despite these improvements, persistent deficiencies remained in both groups, especially in checking breathing/pulse during neonatal resuscitation and properly removing and disinfecting aprons in infection prevention practice.

The findings of this study demonstrate that midwifery graduates trained under a competency-based curriculum performed significantly better than their counterparts in conventional programs in both neonatal resuscitation and infection

prevention. Specifically, CBCGs achieved a 15% higher mean score in neonatal resuscitation performance, highlighting the effectiveness of competency-based training in enhancing critical clinical skills. Improvements were observed in tasks such as airway suctioning and chest rise assessment, which are essential for timely and effective resuscitation of newborns with birth asphyxia. This suggests that the incorporation of simulation-based training and early clinical exposure in the revised curriculum may have contributed to better skill acquisition and retention in emergency newborn care.

Similarly, infection prevention practices were more consistently implemented by CBCGs, who scored nearly 10 percentage points higher than CCGs. This performance gap was particularly evident in fundamental areas that are critical for preventing maternal and neonatal infections during labor and delivery. These findings align with the existing literature, which indicates that competency-based education, when designed with deliberate practice and clinical integration, can substantially improve students' procedural adherence to evidence-based guidelines.

Despite overall improvements among CBCGs, persistent deficiencies were observed in both groups, particularly in critical steps such as checking breathing/pulse during neonatal resuscitation and proper removal and disinfection of aprons. These gaps may result from limited opportunities for repeated hands-on practice during training, retention challenges over time, and variability in clinical supervision. Additionally, graduates' performance may also be influenced by post-graduation work environments, such as staffing levels, equipment availability, and supervisory support, which may contribute to observed skill deficiencies, suggesting that curriculum effectiveness is influenced by the workplace context.

The findings of this study demonstrate that midwifery graduates trained under CBCGs performed significantly better than those trained under CCGs in both neonatal resuscitation and infection prevention. This aligns with previous evidence indicating that competency-based curricula, when integrated with simulation and early clinical exposure, enhance skill acquisition and performance in emergency care procedures [11,12].

Studies in LMICs found that midwives trained with simulation and structured clinical mentorship performed significantly better in neonatal resuscitation steps compared to those without such exposure [13]. Similarly, frequent hands-on training and competency assessments improved neonatal care outcomes among midwives [13]. These reinforce the importance of competency-based education in equipping midwives with the necessary skills.

Infection prevention practices were also higher among CBCGs, who performed 76.9% of the required tasks compared to 67.0% by CCGs, resulting in an effect size of 9.9%. This is consistent with findings from Ethiopia and other LMICs, where curricula emphasizing active learning and repeated clinical practice have been associated with improved adherence to infection prevention protocols [14]. In particular, hand hygiene, PPE use, and decontamination procedures were better implemented among CBCGs, like a study in Nepal where the integration of infection control modules and facility-based simulation improved maternal and newborn infection outcomes [15].

Despite these gains, performance gaps persisted in both groups. These results mirror previous studies from Ethiopia and Uganda, which highlighted that gaps in procedural adherence often persist despite curriculum reforms, especially when clinical supervision and reinforcement mechanisms are weak [12,16].

Overall, these findings support the global evidence that competency-based education enhances clinical performance in maternal and newborn care. However, they also emphasize the need for comprehensive implementation strategies, including strengthened mentorship, routine skills reinforcement, and alignment between theoretical instruction and hands-on practice. Addressing these elements will be critical to fully realizing the potential of curriculum reform in improving health outcomes.

**Limitations.** Selection bias was minimized by including BSc midwifery graduates from third-generation Ethiopian universities with similar establishment years, resources, and support. All participants had less than one year of service, had not received training in neonatal resuscitation and infection prevention, and had passed the national licensure exam, thereby reducing variability due to experience or post-graduation learning.

Measurement bias was reduced by using a validated observation tool and employing two trained, independent observers whose average scores were used. Blinding data collectors limited observer bias to participants' group assignments.

The Hawthorne effect was addressed by ensuring confidentiality, omitting provider and facility identifiers, and presenting the study as a process-focused rather than evaluative one. Some residual performance changes may have remained, but they likely affect both groups equally.

While simulation-based assessment provides a controlled environment to evaluate neonatal resuscitation skills, it may not fully capture performance under real-life clinical pressure. Graduates may demonstrate higher performance in simulation than in actual labor settings, or conversely, may underperform due to unfamiliarity with the simulated environment. Thus, results should be interpreted with caution, recognizing that simulation outcomes may overestimate or underestimate true clinical competence.

**Generalizability.** Conducted across diverse settings using validated tools and trained raters, the study's findings are likely generalizable to other LMICs. The higher performance of CBCGs in both lifesaving procedures (neonatal resuscitation and infection prevention) drives the broader adoption of competency-based curricula.

**Suggestions.** To address the remaining gaps in clinical performance, midwifery education programs should strengthen the implementation of competency-based approaches by incorporating more frequent simulation exercises, structured clinical mentorship, and targeted feedback mechanisms. Ensuring a strong alignment between theoretical content and hands-on practice is essential. Additionally, periodic refresher training should be considered to maintain and enhance clinical skills over time.

**Policy and practice implications.** These findings have important implications for national policy and midwifery practice. The higher performance of CBCGs in neonatal resuscitation and infection prevention underscores the value of simulation-based training, early clinical exposure, and structured mentorship. Results can guide curriculum reform by targeting persistent skill gaps, inform continuing professional development through refresher training and competency-focused workshops, and support accreditation and quality assurance by integrating measurable clinical competencies into program standards, ensuring midwifery graduates are equipped to deliver high-quality maternal and newborn care nationwide.

## 5. Conclusion

This study demonstrates that competency-based midwifery education can improve graduates' performance in neonatal resuscitation and infection prevention relative to conventional curricula. However, the persistence of skill gaps in both groups highlights that curriculum effectiveness is not absolute and may depend on the quality of implementation, mentorship, and reinforcement of practice. These findings frame a testable and debatable argument: while competency-based curricula have the potential to enhance clinical competence, ongoing evaluation, targeted improvements, and context-specific adaptations are essential to fully realize their effectiveness in improving maternal and neonatal outcomes.

## Supporting information

**S1 Data. Data set.**
(SAV)

**S1 Fig. Study workflow and procedures for delivery observations.**
(DOCX)

**S1 Table. Background of clients observed and characteristics of health facilities.**
(DOCX)

## Acknowledgments

We sincerely thank all study participants, including the service providers, women observed during labor and delivery, and the health facilities for their generous cooperation. We also express our deep gratitude to the Center for International Health at LMU, LMU University Hospital, and Ludwig-Maximilians-Universität of Munich, Germany, for their support of this research project.

## Author contributions

**Conceptualization:** Awoke Giletew Wondie, Matthias Siebeck, Tegbar Yigzaw Sendekie, Martin Fischer.

**Data curation:** Awoke Giletew Wondie.

**Formal analysis:** Awoke Giletew Wondie, Matthias Siebeck, Tegbar Yigzaw Sendekie, Martin Fischer, Markus Berndt.

**Funding acquisition:** Awoke Giletew Wondie, Tegbar Yigzaw Sendekie.

**Investigation:** Awoke Giletew Wondie.

**Methodology:** Awoke Giletew Wondie, Matthias Siebeck, Tegbar Yigzaw Sendekie.

**Project administration:** Awoke Giletew Wondie, Tegbar Yigzaw Sendekie.

**Resources:** Awoke Giletew Wondie.

**Software:** Awoke Giletew Wondie.

**Supervision:** Awoke Giletew Wondie, Matthias Siebeck, Tegbar Yigzaw Sendekie, Martin Fischer, Markus Berndt.

**Validation:** Awoke Giletew Wondie.

**Visualization:** Awoke Giletew Wondie, Matthias Siebeck, Tegbar Yigzaw Sendekie, Martin Fischer, Markus Berndt.

**Writing – original draft:** Awoke Giletew Wondie.

**Writing – review & editing:** Awoke Giletew Wondie, Matthias Siebeck, Tegbar Yigzaw Sendekie, Martin Fischer, Markus Berndt.

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
