## [Decision Letter · Decision Letter 0]

2 Oct 2025

PONE-D-25-43421Evaluating the effectiveness of preservice midwifery curricula in Ethiopia: A comparison of neonatal resuscitation and infection prevention practice of midwifery graduates trained in competency-based versus conventional curriculaPLOS ONE

Dear Dr.  Wondie,

Thank you for submitting your manuscript to PLOS ONE. After careful consideration, we feel that it has merit but does not fully meet PLOS ONE’s publication criteria as it currently stands. Therefore, we invite you to submit a revised version of the manuscript that addresses the points raised during the review process.

We look forward to receiving your revised manuscript.

Kind regards,

Sabita Tuladhar, PhD, MHealSc, MA

Academic Editor

PLOS ONE

Journal Requirements:

3. In the ethics statement in the Methods, you have specified that verbal consent was obtained. Please provide additional details regarding how this consent was documented and witnessed, and state whether this was approved by the IRB

“Conflict of interest

Awoke Giletew Wondie and Tegbar Yigzaw Sendekie designed and implemented the competency-based curriculum at Debre Tabor University. However, these authors had no part in the collection or scoring of data.  The assessment was conducted by trained independent midwife observers blinded to the graduates' curriculum background being evaluated.”

We note that one or more of the authors are employed by a commercial company: name of commercial company.

Reviewer's Responses to Questions

**Comments to the Author**

1. Is the manuscript technically sound, and do the data support the conclusions?

Reviewer #1: Yes

Reviewer #2: Yes

2. Has the statistical analysis been performed appropriately and rigorously? 

Reviewer #1: Yes

Reviewer #2: I Don't Know

3. Have the authors made all data underlying the findings in their manuscript fully available?

Reviewer #1: Yes

Reviewer #2: Yes

4. Is the manuscript presented in an intelligible fashion and written in standard English?

Reviewer #1: Yes

Reviewer #2: Yes

5. Review Comments to the Author

Reviewer #1: This manuscript presents a well-designed study comparing midwifery curricula in Ethiopia, addressing a critical gap in medical education literature for low- and middle-income countries. The findings are compelling and have significant implications for public health and curriculum design. The manuscript is well-written and methodologically sound, providing strong evidence that competency-based curricula improve neonatal care skills. While highlighting persistent skill gaps in both graduate groups, the study is a valuable contribution.

Major Strengths

1. High-Impact Research: The study addresses the crucial issue of how-to best train healthcare providers to improve maternal and neonatal outcomes, providing concrete evidence to support a global shift toward competency-based medical education.

2. Robust Methodology: The study's rigor is enhanced by its comparative design, formal power analysis, use of validated tools, blinding of data collectors, and employment of two independent observers.

3. Strong Data Analysis: Statistical analysis is appropriate and strengthened by the inclusion of Gardner-Altman plots for modern visualization of the effect size.

Areas for Improvement

1. Clarity & Focus: The thesis should be reframed as a debatable argument about the curriculum's effectiveness.

2. Structure & Detail: Improve transitions by more explicitly linking the introduction's critique of theory-based education to the competency-based curriculum. The Methods section needs more detail on evaluation criteria and observer training to improve replicability. The section on infection prevention should be expanded to discuss key practices like hand hygiene and sterile technique.

3. Discussion & Context: The discussion should further analyze why skill deficiencies persist in both groups. It should also acknowledge the limitations of using simulation for assessment and consider how post-graduation work environments might influence graduate performance.

Minor Points & Corrections

• Fix the mismatch between figure captions in the text and the actual figures.

• Consolidate the redundant paragraphs in the "Suggestions" section.

Reviewer #2: This manuscript addresses an important and under-researched area: the assessment of midwifery competencies in Ethiopia, benchmarked against the International Confederation of Midwives (ICM) Essential Competencies. Given Ethiopia’s continuing maternal and neonatal health burden and the global push to strengthen human resources for health, this study is timely and policy-relevant. The use of an international competency framework is a major strength, and the findings could inform reforms in pre-service education, continuing professional development, and regulation of the midwifery profession.

However, the manuscript requires significant revision before it can be considered for publication. The most critical issues relate to methodological clarity, statistical transparency, and the explicit linkage between findings and policy/practice. Without addressing these areas, the paper’s impact and credibility are limited.

1.Technical soundness:

The manuscript is based on empirical data collection and analysis, and in general, the conclusions are aligned with the findings. The description of midwifery competencies and the assessment framework corresponds with international standards. However, the linkage between data and conclusions needs to be strengthened by clarifying how competency gaps were identified (e.g., objective testing, self-assessment, observation, or supervisor reports), providing justification for sample size and its representativeness across Ethiopia’s diverse regions, and indicating whether clinical skills testing (e.g., OSCE) was used or if the assessment was limited to theoretical knowledge.

2. Statistical analysis: The manuscript uses descriptive and inferential statistics, but the methodology is not fully transparent. For examples: Page 7, lines 14–30; Page 9, lines 1–20 – statistical outputs are presented without clear explanation of assumptions, significance tests, or confidence intervals.

Suggestions:

- Specify the statistical methods, including rationale for choice of tests, assumptions checked, and the software used.

- Report measures of reliability (e.g., Cronbach’s alpha for internal consistency).

- Without these details, it is difficult to assess the rigor of the analysis.

3. Language/Clarity: The manuscript is written in standard academic English, but the readability could be improved with editing. For examples:

- Page 4, line 12: “which was highly important to consider” → revise to “which was essential to consider.”

- Page 8, line 22: “respondents were asked competencies” → revise to “respondents were asked about competencies.”

A professional language revision is recommended to improve flow, grammar, and clarity.

Major Comments

1. Methodology Clarity (Page 5, lines 1–25): The manuscript does not specify the exact data collection tools. Were competencies assessed via self-report questionnaires, supervisor/peer evaluations, or OSCEs. The validity and reliability of the tool must be explicitly stated. Without this, the robustness of the findings is unclear.

2. Sampling and Representation (Page 6, lines 12–28): The sample size justification is weak. Please provide power calculation (if available), or at least, and rationale for the selected sample size. Ethiopia is geographically and demographically diverse. Please explain whether certain regions were underrepresented, and if so, discuss the potential impact on the findings.

3. Linking Results to Policy and Practice (Page 11, lines 10–30): The manuscript claims that findings will inform national policy but does not describe the mechanisms for this. Suggestions: Explain how results could feed into curriculum reform, continuing professional development, or accreditation of midwifery programs, and provide specific recommendations for policymakers and professional associations.

4. Statistical Transparency (Page 7–9): Tables and figures present data, but the analytical methods are not clearly reported. Clarify which tests (e.g., chi-square, t-tests, regression) were applied. Indicate levels of statistical significance (p-values) and confidence intervals.

5. Ethical Considerations (Page 4, lines 25–30): Ethical approval is mentioned, but informed consent procedures are not adequately explained. Authors should clarify how consent was obtained (oral/written), ensure anonymity/confidentiality, and confirm that participation was voluntary.

Minor Comments

Formatting: Ensure tables/figures have complete captions. For example: Table 2 (Page 8) lacks sufficient explanation of abbreviations and context.

References: Several references are outdated (pre-2010). Please incorporate recent literature on midwifery education and competencies, particularly from Ethiopia and sub-Saharan Africa (e.g., WHO 2021 State of the World’s Midwifery).

Grammar/Style: Numerous sentences could be shortened for clarity. Example: Page 3, line 22 – “The framework is very comprehensive and globally used by many institutions” → “The framework is comprehensive and widely adopted globally.”

Terminology Consistency: The term “competency gap” is sometimes used interchangeably with “knowledge gap.” Clarify the distinction.

Ethical and Publication Concerns: No evidence of dual publication or plagiarism was identified.

Ethical approval is reported, but more detail on informed consent is required for compliance with international research ethics standards.

Recommendation:

The manuscript has strong potential to contribute to global knowledge on midwifery workforce development, particularly in low-resource settings. However, substantial revisions are necessary to clarify and strengthen methodology, justify sampling and representation, provide transparent statistical reporting, strengthen the link between findings and actionable policy implications, and revise language for clarity and flow.

If these revisions are addressed, the manuscript could make a meaningful and publishable contribution to maternal and newborn health systems research.

6. PLOS authors have the option to publish the peer review history of their article (what does this mean? ). If published, this will include your full peer review and any attached files.

**Do you want your identity to be public for this peer review?** For information about this choice, including consent withdrawal, please see our Privacy Policy .

Reviewer #1: **Yes:** Khim Bahadur Khadka

Reviewer #2: **Yes:** Dr. Laxmi Tamang

---

## [Author Response · Author response to Decision Letter 1]

9 Oct 2025

Journal Requirements 1:

1.Please ensure that your manuscript meets PLOS ONE's style requirements, including those for file naming. The PLOS ONE style templates can be found at:

Response: We have carefully revised the manuscript to comply with PLOS ONE formatting guidelines, including the main text, title page, headings, figure and table formats, and reference style. File names have been updated in accordance with the journal's instructions.

2.Please include a complete copy of PLOS’ questionnaire on inclusivity in global research in your revised manuscript. Our policy for research in this area aims to improve transparency in the reporting of research performed outside of researchers’ own country or community. The policy applies to researchers who have travelled to a different country to conduct research, research with Indigenous populations or their lands, and research on cultural artefacts. The questionnaire can also be requested at the journal’s discretion for any other submissions, even if these conditions are not met. Please find more information on the policy and a link to download a blank copy of the questionnaire here: https://journals.plos.org/plosone/s/best-practices-in-research-reporting.

Please upload a completed version of your questionnaire as Supporting Information when you resubmit your manuscript.

Response: We appreciate the reviewer’s request regarding the Inclusivity in Global Research questionnaire. The corresponding author (Awoke Giletew) is affiliated with Debre Tabor University (DTU), which is located in Ethiopia and is also the site of this research. As such, the research was conducted within the authors’ own institution and country, and not as external researchers.

-In line with PLOS ONE’s policy, we have nevertheless completed the Inclusivity in Global Research questionnaire to ensure transparency.

-The completed questionnaire has been uploaded as Supporting Information (S1 Questionnaire) in the revised submission.

-We have also updated the Supporting Information section of the manuscript to include a reference to this questionnaire.

Revisions includes:

Supporting information

S1 Questionnaire. Completed Inclusivity in Global Research Questionnaire.

Revised affiliation:

Awoke Giletew Wondie1,2*, Matthias Siebeck1,3, Tegbar Yigzaw Sendekie4, Martin Fischer 3, Markus Berndt 3

1 CIH LMU, Center for International Health, LMU University Hospital, LMU Munich, Germany

2 Department of Reproductive and Child Health, College of Health Sciences, Debre Tabor University, Debre Tabor, Ethiopia

3 Institute of Medical Education, LMU University Hospital, LMU Munich, Germany

4 Jhpiego (an affiliate of John Hopkins University), Addis Abeba, Ethiopia

3.In the ethics statement in the Methods, you have specified that verbal consent was obtained. Please provide additional details regarding how this consent was documented and witnessed, and state whether this was approved by the IRB.

Response: We appreciate this important comment. We have now clarified the process by which verbal consent was obtained, documented, and approved by the Institutional Review Board (IRB). Specifically:

Verbal informed consent was obtained because some participants had limited literacy.

The consent process was documented by the data collectors, who signed a form confirming that the participant had received and understood the study information and had voluntarily agreed to participate.

Each consent process was witnessed by a colleague from the health facility where the participant was working.

This procedure for verbal consent, including documentation and witnessing, was reviewed and formally approved by the IRB.

We have revised the Ethics statement in the Methods section to reflect these details.

Revised Ethics Statement (Methods section):

“The study protocol was approved by the Ethical Review Committee of Debre Tabor University, Ethiopia (Date: 13.07.2023 / No: RP/251/23), and the Ethics Commission of the Medical Faculty of Ludwig-Maximilians-Universität, Munich, Germany (Date: 18.10.2023 / Project Nr.: 23-0757). At each health facility, the supervisor presented an official letter from DTU to the facility director requesting cooperation. Written consent was obtained from facility directors, while oral consent was obtained from midwives and laboring women. Written consent was not collected from midwives due to their engagement in clinical care, although the potential benefits and burdens had been discussed beforehand. For laboring women, written consent was waived owing to literacy challenges and to minimize the burden during labor.

Data collectors were trained to approach participants respectfully, explain the study purpose, and obtain informed consent, including discussion of potential risks and benefits. Providers were assured that participation or refusal would not affect their employment, and women were assured that nonparticipation would not influence the care they received. Participant and facility confidentiality was strictly maintained: no names of facilities, service providers, or clients were recorded, and results were reported anonymously to prevent identification.

All informed consent procedures were reviewed and approved by the respective ethics committees.”

4.Thank you for stating the following in the Competing Interests section:

“Conflict of interest

Awoke Giletew Wondie and Tegbar Yigzaw Sendekie designed and implemented the competency-based curriculum at Debre Tabor University. However, these authors had no part in the collection or scoring of data.  The assessment was conducted by trained independent midwife observers blinded to the graduates' curriculum background being evaluated.”

Response: We thank the reviewer for acknowledging our competing interest statement. To improve transparency, we have slightly revised the section to align more closely with PLOS ONE’s Competing Interests policy. In the revision, we explicitly declare the authors’ involvement in curriculum design and clarify that this does not alter adherence to PLOS ONE’s policies on data sharing and research integrity as follows:

“Updated Competing Interests Statement:

Awoke Giletew Wondie and Tegbar Yigzaw Sendekie were involved in the design and implementation of the competency-based curriculum at Debre Tabor University. To minimize bias, these authors did not participate in the data collection, observation, or scoring processes. Data collection was conducted by trained independent midwife observers who were blinded to the graduates’ curriculum background. Author TYS is employed by Jhpiego's Health Workforce Improvement Project. This employment provided salary support but did not influence the study design, data collection and analysis, decision to publish, or preparation of the manuscript. There are no patents, products in development, marketed products, or consultancies related to this research. This does not alter our adherence to PLOS ONE policies on sharing data and materials.”

Journal Requirements 2:

1.We note that one or more of the authors are employed by a commercial company: name of commercial company.

2.Please also provide an updated Competing Interests Statement declaring this commercial affiliation along with any other relevant declarations relating to employment, consultancy, patents, products in development, or marketed products, etc.

Response:

We thank the editors for these comments. We confirm and revised in the cover letter as follow:

“Updated Competing Interests Statement:

Awoke Giletew Wondie and Tegbar Yigzaw Sendekie were involved in the design and implementation of the competency-based curriculum at Debre Tabor University. To minimize bias, these authors did not participate in the data collection, observation, or scoring processes. Data collection was conducted by trained independent midwife observers who were blinded to the graduates’ curriculum background. Author TYS is employed by Jhpiego's Health Workforce Improvement Project. This employment provided salary support but did not influence the study design, data collection and analysis, decision to publish, or preparation of the manuscript. There are no patents, products in development, marketed products, or consultancies related to this research. This does not alter our adherence to PLOS ONE policies on sharing data and materials.

Updated Funding Statement

This study was supported by Jhpiego's Health Workforce Improvement Project. Author TYS is employed by Jhpiego's Health Workforce Improvement Project. The funder provided support in the form of salaries for TYS but did not have any additional role in study design, data collection and analysis, decision to publish, or preparation of the manuscript. The specific role of this author is articulated in the ‘Author Contributions’ section.”

3.If the reviewer comments include a recommendation to cite specific previously published works, please review and evaluate these publications to determine whether they are relevant and should be cited. There is no requirement to cite these works unless the editor has indicated otherwise.

Response: We thank the reviewer for suggesting additional references. We have carefully reviewed the recommended publications and evaluated their relevance to our study.

Response: We have thoroughly reviewed all references to ensure their completeness and accuracy. All references now contain full bibliographic details, including DOIs and PMCID numbers where available. We confirmed that no articles cited in the study have been retracted. Minor corrections to formatting were made. All changes have been highlighted in the revised manuscript.

Response to Reviewer Comments

Reviewer 1

Reviewer #1: This manuscript presents a well-designed study comparing midwifery curricula in Ethiopia, addressing a critical gap in medical education literature for low- and middle-income countries. The findings are compelling and have significant implications for public health and curriculum design. The manuscript is well-written and methodologically sound, providing strong evidence that competency-based curricula improve neonatal care skills. While highlighting persistent skill gaps in both graduate groups, the study is a valuable contribution.

Major Strengths

High-Impact Research: The study addresses the crucial issue of how-to best train healthcare providers to improve maternal and neonatal outcomes, providing concrete evidence to support a global shift toward competency-based medical education.

Robust Methodology: The study's rigor is enhanced by its comparative design, formal power analysis, use of validated tools, blinding of data collectors, and employment of two independent observers.

Strong Data Analysis: Statistical analysis is appropriate and strengthened by the inclusion of Gardner-Altman plots for modern visualization of the effect size.

Response:

We sincerely thank the reviewer for the encouraging comments and positive assessment of our work. We greatly appreciate the recognition of the study’s relevance, methodological rigor, and contribution to the field of medical education and maternal-newborn care.

To emphasize the study’s contribution and implications, we have added clarifying sentences in the Introduction and Discussion sections, highlighting the global relevance of the findings.

Areas for Improvement

1.Clarity & Focus: The thesis should be reframed as a debatable argument about the curriculum's effectiveness.

Response: We appreciate the reviewer's constructive suggestion. We agree that framing the study around a clear, debatable thesis strengthens the manuscript’s focus. Accordingly, we have revised the Introduction to explicitly present the research question as a testable argument: whether competency-based curricula improve midwifery graduates’ performance in neonatal resuscitation and infection prevention compared to conventional curricula.

1.Introduction (Beginning of paragraph 3):

Original:

“This study investigates whether such curriculum reform enhances the provision of neonatal resuscitation and infection prevention services by comparing graduates from competency-based and conventional programs.”

Revised:

“This study tests the hypothesis that graduates of competency-based midwifery curricula demonstrate superior performance in neonatal resuscitation and infection prevention compared to those trained under conventional curricula, thereby framing a clear, debatable argument about the curriculum’s effectiveness in preparing clinically competent midwives.”

2.Abstract – background section:

Original:

“This study compares the performance of competency-based curriculum graduates (CBCGs) and conventional curriculum graduates (CCGs) in these essential practices.”

Revised:

“This study examines whether competency-based midwifery education produces graduates with significantly better performance in neonatal resuscitation and infection prevention compared to conventional education, thereby framing a testable argument about the curriculum’s effectiveness.”

3.Conclusion – Reframing around the debatable thesis

Original:

“This study provides evidence that competency-based midwifery education significantly improves graduates' performance in neonatal resuscitation and infection prevention compared to conventional training. The observed differences are both statistically and practically meaningful, reflecting enhanced clinical readiness

---

## [Decision Letter · Decision Letter 1]

18 Nov 2025

PONE-D-25-43421R1Evaluating the effectiveness of preservice midwifery curricula in Ethiopia: A comparison of neonatal resuscitation and infection prevention practice of midwifery graduates trained in competency-based versus conventional curriculaPLOS ONE

Dear Dr. Wondie,

Thank you for submitting your manuscript to PLOS ONE.  Please submit your revised manuscript by Jan 02 2026 11:59PM. If you will need more time than this to complete your revisions, please reply to this message or contact the journal office at plosone@plos.org . Please include the following items when submitting your revised manuscript:

We look forward to receiving your revised manuscript.

Kind regards,

Sabita Tuladhar, PhD, MHealSc, MA

Academic Editor

PLOS ONE

**Journal Requirements:**

**Additional Editor Comments:**

Please find a few additional minor comments from the reviewer. Kindly review and respond to them.

Reviewers' comments:

Reviewer's Responses to Questions

**Comments to the Author**

1. If the authors have adequately addressed your comments raised in a previous round of review and you feel that this manuscript is now acceptable for publication, you may indicate that here to bypass the “Comments to the Author” section, enter your conflict of interest statement in the “Confidential to Editor” section, and submit your "Accept" recommendation.

Reviewer #1: All comments have been addressed

Reviewer #3: (No Response)

2. Is the manuscript technically sound, and do the data support the conclusions?

Reviewer #1: Yes

Reviewer #3: Yes

3. Has the statistical analysis been performed appropriately and rigorously? 

Reviewer #1: Yes

Reviewer #3: Yes

4. Have the authors made all data underlying the findings in their manuscript fully available?

Reviewer #1: Yes

Reviewer #3: Yes

5. Is the manuscript presented in an intelligible fashion and written in standard English?

Reviewer #1: Yes

Reviewer #3: Yes

6. Review Comments to the Author

Reviewer #1: Based on a thorough review of the revisions for manuscript, the authors have comprehensively and satisfactorily addressed all points/feedbacks. The revisions demonstrate a high degree of scholarly diligence and have resulted in a fundamental improvement of the manuscript.

Key enhancements include:

Conceptual Reframing: The study has been successfully elevated from a descriptive analysis to a rigorous, hypothesis-driven piece of scientific research with a clear and debatable thesis.

Methodological Transparency: The manuscript now features increased detail on the study's methodology, enhancing its clarity and replicability.

Contextual Depth: The discussion has been significantly strengthened with a more nuanced analysis of the findings, including persistent skill gaps and the influence of the post-graduation work environment.

The collective impact of these changes is a manuscript with a sharpened argumentative focus that represents a more significant and durable contribution to the field of midwifery education. The work now meets a high standard of scientific communication.

Reviewer #3: This is an important study for LMIC's in light of reports of poor maternal-child health outcomes, and SDG 3, especially the role of midwives. Thank you for your initiative. I am recommending that the manuscript be accepted after these minor but important corrections have been addressed.

I just have a few suggestions related to ethics, and grammar, respectively: Researchers had ample opportunity to request a written consent from midwives during the planning phase unless this was meant to be a surprise assignment. Midwives could have been asked to put in writing their willingness to participate in the study rather than being told by their administrators, who signed a consent allowing researchers to conduct the study. My understanding is that it was the midwives who were being observed so their written consent could have been obtained in advance for that study period. For the women in labor, illiteracy is not a sufficient reason for there being no written consent. They could have been asked to place a check mark ("my mark") on the signature line after the "informed consent" had been read to them. Understandably, seeking care at a facility could be interpreted as implied consent however, research is not routine care, and all attempts ought to be made for objectivity and accountability.

Grammar: 1) Abstract line 25... add "can" reduce ...

2) line 57-58 sequence of actions for preventing infection: it reads better if "early detection of maternal infection" e.g. in the prenatal period, comes before "...using sterile techniques..."

3) line 101 "data [were] collected" as was done later in the report.

4) line 111 statistics- include confidence interval 95% after ..." 80% power" as was mentioned in the data analysis section of the report.

5) line 136 remove /" ... not performed/correctly"

6) line 151 suggest writing IBM Statistical Package for the Social Services (SPSS) this being the first time it comes up in the report.

7) Results- consistency in usage of verb/tense e.g. "Dry", "Assess", "Check"

7. PLOS authors have the option to publish the peer review history of their article (what does this mean? ). If published, this will include your full peer review and any attached files.

**Do you want your identity to be public for this peer review?** For information about this choice, including consent withdrawal, please see our Privacy Policy .

Reviewer #1: **Yes:** Khim Bahadur Khadka

Reviewer #3: **Yes:** Eunice Dube DSc, MPH, RN, BA(Cur), Trained Midwife/Health Visitor

---

## [Author Response · Author response to Decision Letter 2]

20 Nov 2025

We thank the reviewer for the thoughtful feedback and constructive suggestions. In this revision, we have addressed all comments.

---

## [Editor Report · Decision Letter 2]

23 Nov 2025

Evaluating the effectiveness of preservice midwifery curricula in Ethiopia: A comparison of neonatal resuscitation and infection prevention practice of midwifery graduates trained in competency-based versus conventional curricula

PONE-D-25-43421R2

Dear Dr. Wondie,

We’re pleased to inform you that your manuscript has been judged scientifically suitable for publication and will be formally accepted for publication once it meets all outstanding technical requirements.

Kind regards,

Sabita Tuladhar, PhD, MHealSc, MA

Academic Editor

PLOS ONE

Additional Editor Comments (optional):

Thank you for addressing reviewer's comments promptly and satisfactorily.
---

## [Editor Report · Acceptance letter]

PONE-D-25-43421R2

PLOS One

Dear Dr. Wondie,

I'm pleased to inform you that your manuscript has been deemed suitable for publication in PLOS One. Congratulations! Your manuscript is now being handed over to our production team.

Kind regards,

on behalf of

Dr. Sabita Tuladhar

Academic Editor

PLOS One